# Gadolinium-enhanced MRI visualizing backflow at increasing intra-renal pressure in a porcine model

**Søren Kissow Lildal**[1]*, **Esben Søvsø Szocska Hansen**[2], **Christoffer Laustsen**[2], **Rikke Nørregaard**[3], **Lotte Bonde Bertelsen**[2,3], **Kirsten Madsen**[4], **Camilla W. Rasmussen**[2], **Palle Jörn Sloth Osther**[5,6], **Helene Jung**[5,6]

1 Department of Urology, Aarhus University Hospital, Aarhus, Denmark, 2 Department of Clinical Medicine, MR Research Centre, Aarhus University, Aarhus, Denmark, 3 Department of Clinical Medicine, Aarhus University, Aarhus, Denmark, 4 Department of Clinical Pathology, Odense University Hospital, Odense, Denmark, 5 Department of Regional Health Research, University of Southern Denmark, Odense, Denmark, 6 Department of Urology, Vejle Hospital–a part of Lillebaelt Hospital, University Hospital of Southern Denmark, Vejle, Denmark

* soerlild@rm.dk

**Data Availability Statement:** All relevant data are within the paper and its Supporting Information files.

## Abstract

### Introduction

Intrarenal backflow (IRB) is known to occur at increased intrarenal pressure (IRP). Irrigation during ureteroscopy increases IRP. Complications such as sepsis is more frequent after prolonged high-pressure ureteroscopy. We evaluated a new method to document and visualize intrarenal backflow as a function of IRP and time in a pig model.

### Methods

Studies were performed on five female pigs. A ureteral catheter was placed in the renal pelvis and connected to a Gadolinium/ saline solution 3 ml/L for irrigation. An occlusion balloon-catheter was left inflated at the uretero-pelvic junction and connected to a pressure monitor. Irrigation was successively regulated to maintain steady IRP levels at 10, 20, 30, 40 and 50 mmHg. MRI of the kidneys was performed at 5-minute intervals. PCR and immunoassay analyses were executed on the harvested kidneys to detect potential changes in inflammatory markers.

### Results

MRI showed backflow of Gadolinium into the kidney cortex in all cases. The mean time to first visual damage was 15 minutes and the mean registered pressure at first visual damage was 21 mmHg. On the final MRI the mean percentage of IRB affected kidney was 66% after irrigation with a mean maximum pressure of 43 mmHg for a mean duration of 70 minutes. Immunoassay analyses showed increased MCP-1 mRNA expression in the treated kidneys compared to contralateral control kidneys.

**Funding:** The authors received no specific funding for this work.

**Competing interests:** The authors have declared that no competing interests exist.

## Conclusions

Gadolinium enhanced MRI provided detailed information about IRB that has not previously been documented. IRB occurs at even very low pressures, and these findings are in conflict with the general consensus that keeping IRP below 30–35 mmHg eliminates the risk of post-operative infection and sepsis. Moreover, the level of IRB was documented to be a function of both IRP and time. The results of this study emphasize the importance of keeping IRP and OR time low during ureteroscopy.

## Introduction

Indications for ureteroscopy have expanded dramatically during recent years, and although ureteroscopy in general is considered a safe procedure, serious complications and even deaths do occur, and these are most often related to sepsis [1]. To understand this, we must focus on how ureteroscopy may push upper urinary tract physiology to pathophysiology [3, 10]. When advancing a ureteroscope to the upper urinary tract and using saline irrigation, which is necessary for vision, the intrarenal pressure will increase significantly [5, 6], and when exceeding certain thresholds, intrarenal backflow may occur: backflow of urine and irrigation fluid to the renal tubules and the venous system. We have studied intrarenal pressure during ureteroscopy quite extensively during recent years [3, 6, 10]; however, our understanding on the relation between intrarenal pressure (IRP) and intrarenal backflow (IRB), which is the likely event immediately before sepsis, is still deficient; and the objective of this study was to approach this knowledge gap. The specific aim was to evaluate and visualize IRB dynamically as a function of intrarenal pressure and time using MRI in a porcine model with a Gadolinium tracer in the irrigation fluid.

## Materials and methods

### Experimental animals

The animal protocol was approved by The National Animal Experiments Inspectorate (Copenhagen, Denmark). Studies were performed on 5 anaesthetized female pigs weighing 45 kg (Påskehøjgård, Ølsted, Denmark). The pigs were fed a standard diet during breeding. Before the study they had access to water but were fasting 12 hours prior to anaesthesia.

After premedication with azaperone (4mg/kg) and midazolam (4 mg/kg), anaesthesia was induced by propofol (4–20 mg/kg) and maintained with sevoflurane (1.2 MAC) and fentanyl (0.03 mg/kg/h). The pigs were orotracheally intubated and mechanically ventilated (GE Healthcare S5 Avance). Hydration was maintained by administration of saline (9 g/l sodium chloride; 10 ml/kg/h) at a temperature of 37˚C through an ear vein.

A cystoscope was inserted through the urethra into the bladder. A ureteral catheter (Selectip®, Bard Medical, Covington, USA) was placed in the distal part of the ureter and a retrograde pyelography was performed to visualize the anatomy of the upper urinary tract. A guide wire (Sensor®, Boston Scientific, Marlborough, MA, USA) was placed via the ureteral catheter, through the ureter, to the renal pelvis, and the cystoscope was removed.

Over the guide wire, and under fluoroscopic guidance, a dual lumen catheter (Cook Medical, Bloomington, IN, USA) was inserted to the distal ureter and through this a second guidewire was inserted to the renal pelvis and the catheter was removed. Over the first guidewire a regular ureteral catheter was placed in the middle of the renal pelvis and over the second

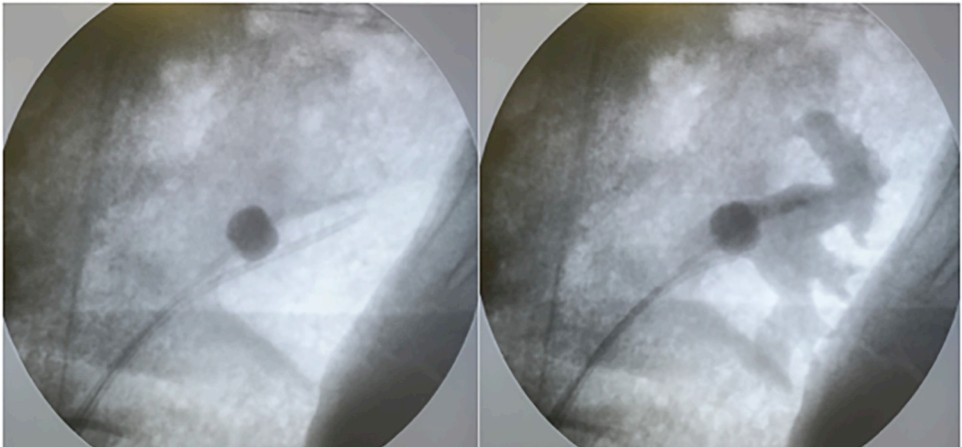

**Fig 1. Pyelography.** Placement of ureteral catheter and balloon catheter for occlusion, irrigation and pressure measurement confirmed by fluoroscopic pyelography.

guidewire an occlusion balloon catheter (Boston Scientific, Marlborough, MA, USA) was inserted and the balloon was inflated just above the level of the uretero-pelvic junction. The correct placement of both catheters was confirmed by contrast fluoroscopic pyelography (Fig 1) and the flow function through both catheters was confirmed by gravity-mediated infusion into the ureteral catheter and free outflow through the occlusion balloon catheter. Both catheters were fixed outside the urinary tract with adhesive tape to the skin of the pig to ensure a fixed position. A urethral catheter was placed in the urinary bladder to ensure drainage of urine and irrigation fluid during the testing period.

The animal was then placed in the MR scanner with the ureteral catheter connected to an irrigation pump and the occlusion catheter to a pressure monitor. A baseline MR scan was done before initiation of irrigation and the baseline intraluminal pressure was registered.

A Gadolinium/saline solution 3 mL/L was mounted in the irrigation pump and irrigation was initiated to evaluate the resulting intraluminal pressure. The flow was continuously regulated to maintain steady intraluminal pressure levels starting with a pressure of 10 mmHg and through the study raising it to 20, 30, 40 and 50 mmHg. At each of these pressure levels, MR scans were performed with 5-minute intervals.

After finishing the MR scan series, the animals were brought back to the OR where both kidneys were removed in vivo, to obtain tissue samples from the upper pole, middle region and lower pole. Samples were preserved in formaldehyde for histopathological evaluation and snap frozen in liquid nitrogen and stored at -80˚C for ELISA and PCR analysis. Finally, the pigs were euthanized under anaesthesia, with 20 ml of pentobarbital, 200mg/ml.

## MR

After initial scout images to locate the position and orientation of the kidneys, for each pressure level, T1-maps of the kidneys were obtained. The images were acquired using a single-shot inversion recovery spin echo sequence in an oblique coronal plane. Seven slices were acquired covering almost the entire kidneys. T1 mapping was performed using inversion time (TI from 200 ms to 1800 ms). The slices thickness was 5 mm, and acquired pixel size was 3 x 3 mm. Repetition time (TR) was 5 secs and echo time (TE) was 20 ms.

Prior to analysis, the images were motion corrected by doing cross correlation of the image gradient maps for the different inversion times. T1-maps were calculated by fitting the pixel

signal to the equation describing the signal at different TI times: $S(TI) = S_0 |(1-2e^{(-TI/T_1)}|$. The development of T1 with increasing pressure was assessed by placing ROI's in different segments of the kidneys, and plotting T1 as function of pressure.

## NGAL and KIM-1 ELISA measurements

ELISA for Neutrophil gelatinase-associated lipocalin (pig NGAL ELISA kit 044, BioPorto) and Kidney Injury Molecule-1 (pig KIM-1/HAVCR1 ELISA kit abx255516, Abbexa) was performed according to the manufacturer´s instructions. In Brief, Kidney tissue was snap-frozen in liquid nitrogen. Thereafter, tissue was homogenized and centrifuged and ELISA performed on the resultant supernatant.

## RNA extraction, cDNA synthesis and quantitative PCR

Total RNA from cortical tissue was isolated using the NucleoSpin RNA II mini kit according to the manufacturers protocol (Macherey Nagel, Nordrhein-Westfalen, Germany). The RNA concentration was measured with spectrophotometry and samples were stored at -80˚C until use.

cDNA synthesis was performed using 0.5 µg RNA using the RevertAid First Strand Synthesis Kit (Thermo Fisher Scientific, MA, USA). For qPCR, 100 ng of cDNA served as a template for the PCR amplification using the Brilliant SYBR® Green qPCR master Mix according to the manufacturer's instruction (Thermo Fisher Scientific, MA, USA) by using an Aria Mx3000P qPCR system (Agilent Technologies, Santa Clara, Ca). Samples and the standards were prepared in duplicate in 96-well plates and PCR was accomplished for 40 cycles consisting of denaturation for 30s at 95˚C followed by annealing and polymerization at 60˚C for 1 min. Emitted fluorescence was detected during the annealing/extention step in each cycle. Specificity was ensured by post-run melting curve analysis and reaction products were separated on agarose gels and imaged. GAPDH was used as housekeeping gene. Primer sequences are shown in Table 1.

## Histology

Kidney tissue was cut in coronal slices though the tip of a papilla and embedded in paraffin. For histological evaluation of kidney morphology, 4 µm thick tissue slices were cut on a microtome and mounted on microscope glass slides (Superfrost). All slides were deparaffinized, and tissue visualized by hematoxylin eosin (H&E) and periodic acid-shiff (PAS) staining. Kidney morphology was evaluated by standard light microscopy (Olympus).

**Table 1. PCR primer sequences.** Primers used for qPCR.

|  | Forward primer | Reverse primer |
|---|---|---|
| **TNF-alpha** | 5'- GGC TGC CTT GGT TCA GAT GT-3 | 5'-CAG GTG GGA GCA ACC TAC AGT T -3' |
| **IL-1beta** | 5'-GAT GAC ACG CCC ACC CTG-3' | 5'-CAA ATC GCT TCT CCA TGT CCC-3' |
| **COX-2** | 5'- CAA AAC CGT ATT GCT GCT GA -3' | 5'- CAA AAC CGT ATT GCT GCT GA -3' |
| **MCP1** | 5'-ACT TGG GCA CAT TGC TTT CCT-3' | 5'-TTT TGT GTT CAC CAT CCT TGC A-3' |
| **IL-6** | 5'-AGA CAA AGC CAC CAC CCC TAA-3' | 5'-CTC GTT CTG TGA CTG CAG CTT ATC-3' |
| **GAPDH** | 5'-AGC AAT GCC TCC TGT ACC AC-3' | 5'-AAG CAG GGA TGA TGT TCT GG-3' |

## Statistics

Biochemical and qPCR was analysed using a two-way mixed-effects model taking into account the multiple samples from the same kidneys and the same animal. Normality was assessed by QQ-plots. All statistical analyses were performed in GraphPad Prism 9 (GraphPad Software, San Diego, CA, USA).

## Results

### MR

The experiment was successfully completed in all 5 animals, and MR clearly showed backflow of Gadolinium into the kidney cortex in all cases. The measured mean baseline pressure before initiation of irrigation was 9.6 mmHg (range: 5–13). Under controlled irrigation the pressure was gradually increased while repeating MR scans until a maximum pressure of approximately 50 mmHg was reached. The mean maximum pressure recorded for all cases at termination of irrigation was however only 43 mmHg due to inability to increase the pressure to more than 20 mmHg in one of the animals, which was probably because of compromised occlusion by the balloon catheter and thus leakage of irrigation fluid down the ureter. We did however in this case find MR changes in the kidney cortex similar to all other cases regarding the degree of backflow and development over time.

Uniformly for all cases, MR showed early signs of intrarenal backflow into the cortex of the kidneys clearly visualised as pyramid shaped changes first appearing in the upper and lower poles (Fig 2). Subsequently more and more similar shaped changes appeared in all other regions until large areas (mean 66% of the renal cortex, Figs 3 and 4) were affected at the end MR scan (Fig 5).

The mean time to first visual renal cortex changes was 15 minutes (range: 5–25 min) and more importantly the mean registered pressure at first visual change was only 21 mmHg (range: 16–25 mmHg).

In order to relate these results to a clinical situation we combined these two factors and calculated the accumulated pressure applied within the pelvis over time (mmHg x min). The mean total time from irrigation start to finish was 70 minutes (range 60–80 min) and the mean total accumulated pressure was 405 mmHg x minutes. The mean accumulated pressure at first visual damage was 70 mmHg x minutes (Fig 6).

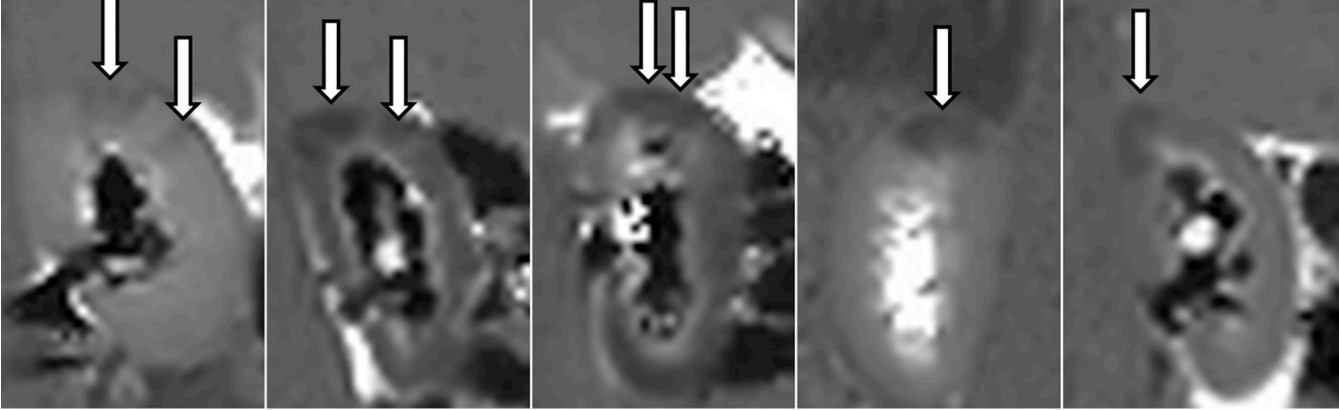

**Fig 2. First visual changes.** First MR-scan for each of the 5 animals with visual changes appearing as dark areas (arrows). In animal no. 4 the study was performed on the right kidney due to catheter placement failure on the left side. 1: 22 min—25 mmHg 2: 5 min– 20 mmHg 3: 25 min– 22 mmHg 4: 20 min– 22 mmHg 5: 5 min– 16 mmHg.

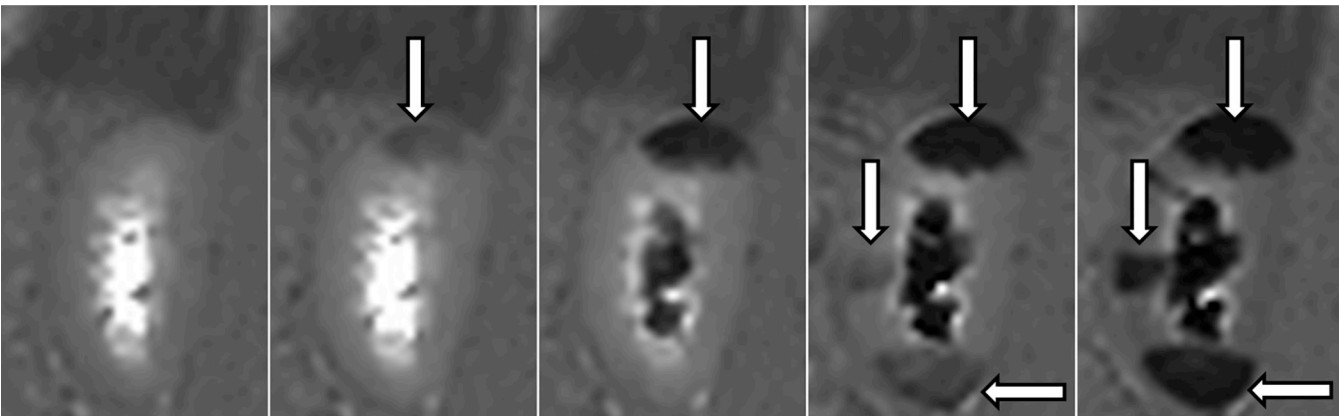

**Fig 3. Time series MR.** Time series of MR-scans in 1 animal from baseline (0 mins) to end scan (80 mins) showing increased T1 values with rising intraluminal pressure (10 mmhg– 50 mmHg) in the right kidney. 0 min—10 mmHg 20 min– 22 mmHg 40 min– 31 mmHg 60 min– 40 mmHg 80 min– 50 mmHg.

On the final MR scan of all cases the areas showing backflow by changes in T1 values were mapped and a fraction of affected/total kidney tissue was calculated. The consequent mean percentage of affected kidney was 66% (range 43.2% - 79.6%) after irrigation with a mean maximum pressure of 43 mmHg for a mean duration of 70 minutes.

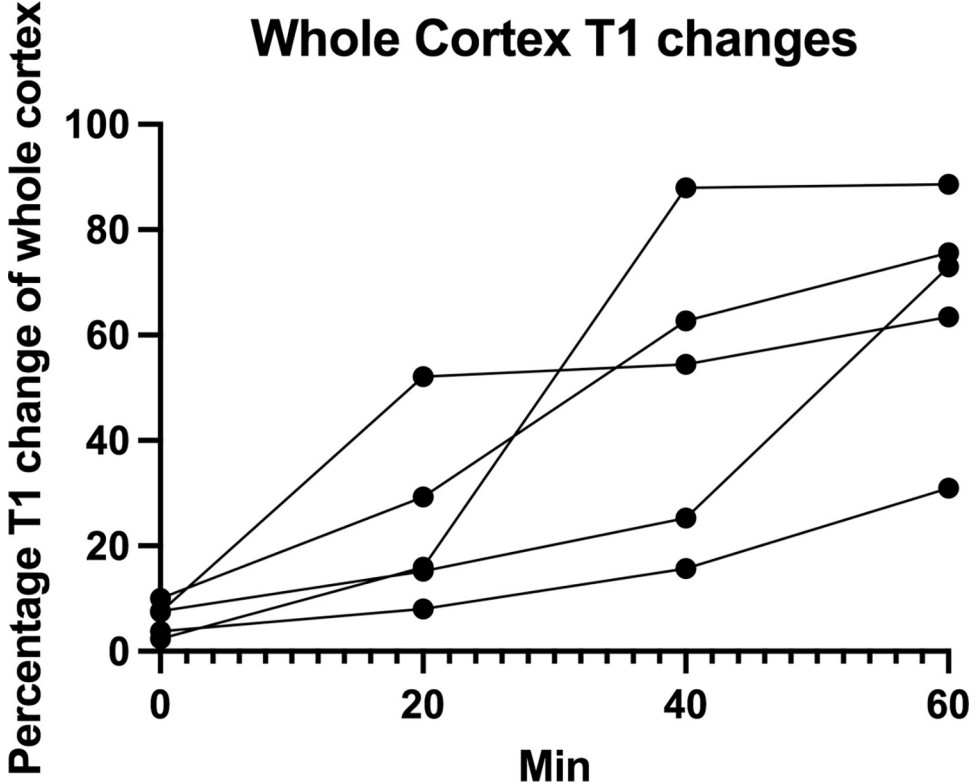

**Fig 4. Whole cortex changes.** Fraction of whole cortex T1 changes as a function of time from baseline scan to 60 mins. scan for all 5 experimental animals. The curves are ended at 60 mins. as some of the experiments were ended at this time point. At 60 minutes a mean of 66% of the cortex was affected by intrarenal backflow.

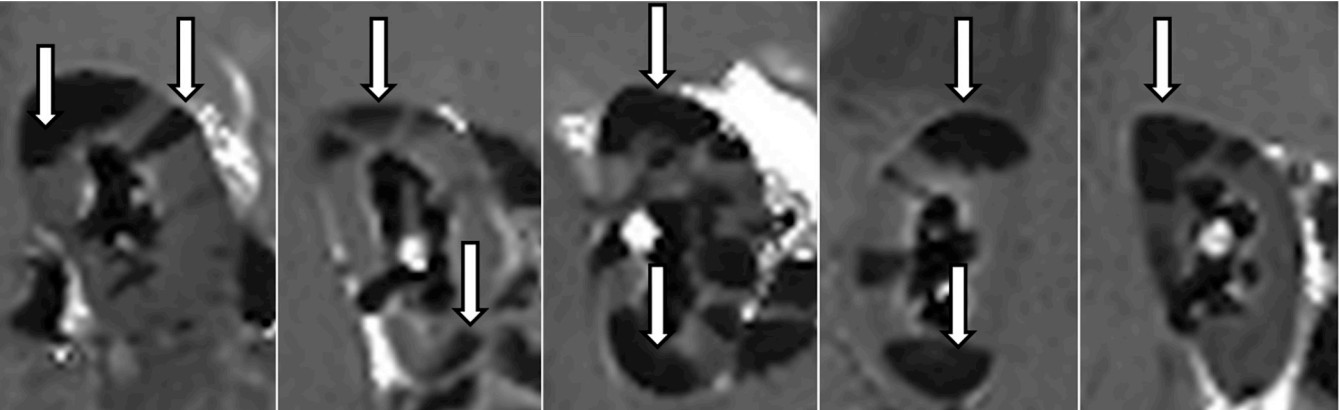

**Fig 5. Final MR scan.** Final MR-scan for the 5 animals showing widespread T1 changes as dark areas (arrows) due to increased intraluminal pressure in the treated kidneys. Time range 60–80 mins; pressure range 20–54 mmHg. In animal no. 4 the study was performed on the right kidney due to ureteral catheter placement failure on the left side. 1: 72 min—51 mmHg 2: 60 min– 20 mmHg 3: 80 min– 40 mmHg 4: 80 min– 50 mmHg 5: 60 min– 54 mmHg.

## Histopathology

The obtained samples from the upper, mid and lower part of the kidneys were evaluated by light microscopy. No histological signs of inflammation or acute tissue damage could be found.

## PCR and immunoassay

Cortical tissue samples were divided into upper, mid and lower pole of the kidney. NGAL and KIM-1 protein levels were analyzed by ELISA analysis and no significant difference was observed between the ureteroscopically treated kidney and the contralateral control kidney. mRNA expression of the inflammation markers COX-2, IL-6, IL-1-beta, MCP 1 and TNF-alfa was evaluated by QPCR analysis. Data showed increased MCP-1 mRNA expression in the

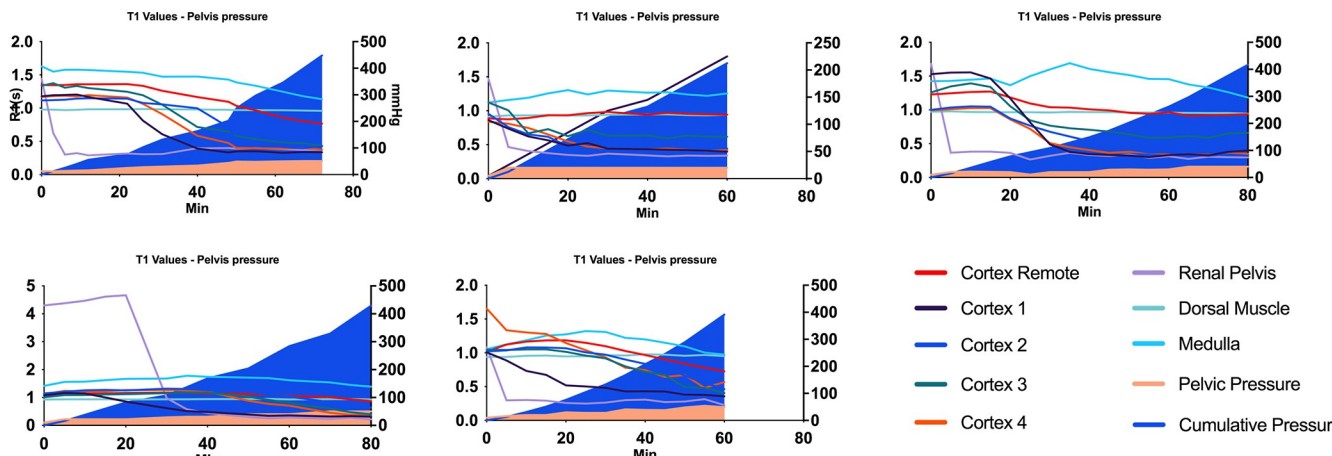

**Fig 6. T1 vs accumulated pelvis pressure.** The development of T1 with increasing accumulated pressure (mmHg x min) was assessed by placing ROI's in different segments of the kidneys, and plotting T1 as function of time x pressure. Left vertical axis: T1; right vertical axis: accumulated pressure; horizontal axis: time. Each graph represents the treated kidney of each of the 5 animals. Cortex remote: visibly non-affected area; cortex 1–4: visibly affected cortical areas; renal pelvis; medulla and dorsal muscle ROIs depicted by lines. Pressure at a given time depicted by colour filled area under the curve. T1 values in affected areas decrease as a function of increasing pressure/time.

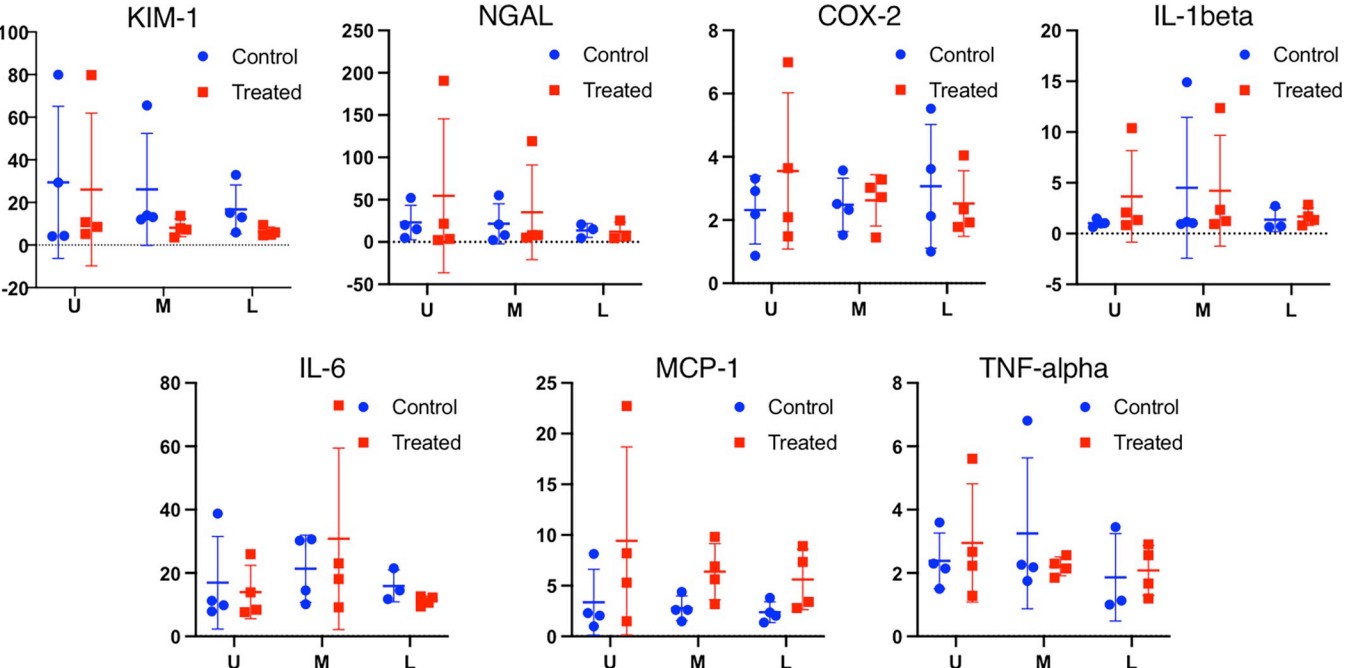

**Fig 7. PCR and ELISA results.** KIM-1 was not associated with any statical significant difference between the upper (U), mid (M) and lower (P) pole of the kidney (p = 0.044), nor between the ureteroscopically treated and the contralateral kidney (p = 0.18). No statistical significant difference was found between the kidney regions and between the contralateral and the ureteroscopic treated kidney respectively for NGAL (p = 0.38, p = 0.45), COX-2 (p = 0.75,p = 0.75), IL-1beta (p = 0.52,p = 0.42), IL-6 (p = 0.33,p = 0.80) and TNF-alpha (p = 0.65,p = 0.79). MCP-1 was on the other hand found to be statistical significant upregulated in the ureteroscopic treated (p = 0.04), while no regional difference was found (p = 0.55). Y-axis units: KIM-1 and NGAL: ng/ml. COX-2, IL-1beta, IL-6, MCP-1 and TNF-alfa: mRNA expression (qPCR marker/GAPDH ratio).

treated kidneys compared to contralateral control kidneys. There was no significant change in mRNA expression of the other inflammation markers (Fig 7).

## Discussion

Miniaturization of ureteroscopic instruments combined with increasingly potent laser technology for the treatment of kidney stones may require forced, high-pressure irrigation to clear stone dust, blood and debris. As a consequence, IRP rises unequivocally during ureterorenoscopic procedures. The physiological IRP in the un-obstructed human kidney is 0–10 mmHg [2]. Human and porcine renal anatomy and physiology are comparable, and several studies have documented IRP in the porcine and human kidney to be at the same level, which was confirmed in the present study [2, 3]. IRP during ureteroscopic treatment of kidney stones can reach 3–400 mm Hg [4, 5]. Previous studies have suggested IRB to occur at approximately 30 mmHg [6, 7]. Increased IRP and IRB constitutes a major predisposing factor for postoperative complications such as sepsis and hemorrhage [8–11] and the per-operative IRP level and irrigation volume appear to be independent risk factors for the development of systemic inflammatory response syndrome (SIRS).

In recent years there has been increasing focus on IRP during endourological procedures and the potential harmful postoperative effects of IRB. As neither the IRP nor the level of IRB is normally determined in clinical settings, the pathophysiological and pathoanatomical changes that may follow ureteroscopy are not fully understood.

In this study we aimed to visually document IRB as a function of increasing IRP and time, using Gadolinium tracer in the irrigation fluid followed by dynamic MR scans. The setup

allowed MR visualization of the gradual backflow beyond the limits of the kidneys at increasing IRP in a porcine model mimicking a clinical situation.

IRB has in former studies been documented using light microscopy, x-ray, ink-irrigation or simple measurement of fluid absorption [6, 12–14]. Visualization of IRB by dynamic MR using Gadolinium [15] contrast agents and T1-mapping MRI represents a new methodology, providing more information of the level and the nature of IRB [16]. Gd-enhanced MRI is a well-established clinical tool for a wide range of indications and as such could be easily translated to clinical usage. It should be noted that Gd contrast agents are contraindicated for intravenous use in patients with chronic kidney disease (CKD) but there is no evidence for kidney damage in endoluminal Gd contrast administration [15, 17, 18].

Using this new method, we were able to determine the mean percentage of affected kidney tissue according to IRP and time exposure. The T1 mapping revealed a mean percentage of affected kidney of 66% at a mean IRP of 43 mmHg. We find it remarkable that over half of the kidney tissue was affected by IRB at this relatively low pressure, which is considered to be a realistic pressure level in a clinical setting during ureteroscopy. The MR scans clearly visualized the increasing degree of IRB, not only as a function of increasing IRP, but interestingly also as a function of time at steady IRP as shown in the graph of animal no. 2 in Figs 6 and 8, where the pressure rose to 20 mmHg immediately and remained at this level. Moreover, MR visualization of the renal parenchyma showed in detail at which pressure level the individual renal pyramids were influenced by IRB, uniformly showing early signs of intrarenal backflow into the cortex of the kidneys clearly visualised as pyramid shaped changes first appearing in the upper and lower pole. Subsequently, similar shaped changes appeared in all other regions until large areas were affected at the end of MR scan. This is in accordance with previous findings by other authors using different methodology [12].

In this study, IRB was investigated at pressure levels of 10–50 mm Hg within 70 minutes of continuous irrigation. The pressure levels and time settings were chosen based on former human studies determining the average IRP during ureteroscopic procedures and on the basis of an assumption that the maximum duration of a ureteroscopic procedure is 70–90 minutes [4, 5, 19]. IRP may be much higher than 50 mmHg as documented by for example Jung and

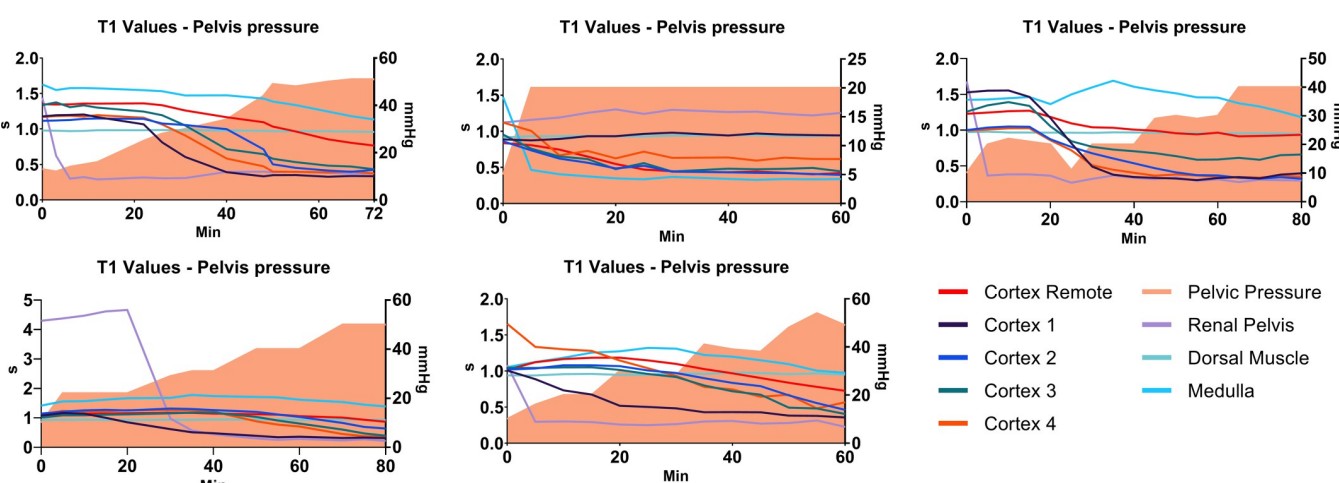

**Fig 8. T1 vs pelvis pressure.** The development of T1 with increasing pressure was assessed by placing ROI's in different segments of the treated kidneys, and plotting T1 as function of time/pressure. Left vertical axis: T1; right vertical axis: pressure; horizontal axis: time. Each graph represents the treated kidney of each of the 5 animals. Cortex remote: visibly non-affected area; cortex 1–4: visibly affected areas; renal pelvis; medulla and dorsal muscle ROIs depicted by lines. Pressure at a given time depicted by colour filled area under the curve. T1 values in affected areas decrease as a function of increasing pressure/time.

Osther and Wilson [5, 4], but IRB was evident in all kidneys within the range of 16–25 mmHg, and consequently we found no indication to examine higher pressure levels in this primary investigation.

Most remarkable, IRB was evidenced at an IRP of only 16 mmHg, range 16–25 mm Hg. To our knowledge, IRB at such low pressures has not been documented earlier. IRB was observed at a mean IRP of 21 mmHg. Recent studies propose IRB to take place at 30–35 mm Hg [6, 13, 20]. Stenberg et al. proposed that the extent of IRB depends on the condition of the kidney: In a hydronephrotic kidney and in kidneys with thin renal cortex, IRB seem to occur at lower IRPs compared to a normal kidney [12]. The kidneys in our study were normal with no signs of hydronephrosis or renal atrophy. The use of Gadolinium MR scan probably identifies signs of IRB and changes in the renal parenchyma in a more accurate way compared to former investigation methods, which may explain our results.

Clinical studies concluding that placement of a ureteral access sheath (UAS) may solve the problems associated with increased IRP by increasing the outflow from the renal pelvis assume that IRB takes place at 30–35 mm Hg. However, IRP during ureteroscopy with an inserted UAS often exceed 25–30 mm Hg [21, 22] and therefore UAS may not always prevent IRB and concomitant postoperative complications. Therefore, these observations raise thoughts about further technological and clinical improvements to secure low-pressure ureteroscopy.

In order to create a measure for accumulated ureteroscopy-induced pressure impact (AUPI) on the kidney, we calculated the accumulated pressure applied within the renal pelvis over time. In a clinical experimental setting, AUPI may be used to compare the accumulated impact (= pressure x time) on the kidney with clinical outcomes such as post-operative infection, fever, sepsis, hospitalization time and biomarker measures. According to other studies, not only the level of IRP, but also operation length has an impact on post-operative outcome [23–25]. We found a mean total time from irrigation start to finish of 70 minutes (range 60–80 min) and a mean total accumulated pressure of 405 mmHg x minutes. The mean AUPI at first visual damage was 70 mmHg x minutes. In future studies, this measure may be used to correlate pressure and OR time and observe changes in IRB accordingly. Until now it is not known if short-lasting, extensive pressure increments exert more damage than long-lasting, moderate pressure. The present study set-up may contribute to answering this question.

The design and setup of this experimental pilot study possess limitations.

In one of the pigs, it was not feasible to increase the pressure to more than 20 mmHg, probably due to leakage of urine from the renal pelvis to the bladder. The pigs varied with respect to ureteral anatomy and diameter, which may explain the inconsistency in pressure recordings. However, the main aim of the study was to obtain a clinical relevant IRP and observe correspondent incidents in the kidney and surroundings at given pressure measurements. It is concluded that these conditions were met in this study design.

The study does not provide information about the potential clinical consequences of the observed IRB. Histological examinations showed no signs of acute tissue damage in the renal parenchyma, which however was not surprising, as parenchymal changes are expected to occur days to weeks after exposure to high pressure according to previous studies [14, 26]. In a future study design, evaluation of renal parenchymal pathology 2–4 weeks after high-pressure exposure would be valuable, which will require post-procedural survival of the pigs.

Biomarker analyses indicate early and mild inflammation. This is in line with the fact that this study is acute and aimed to examine the lower threshold for intrarenal backflow, which was found to be at a very low intrarenal pressure. For the relatively low pressures applied in this study it would not be expected to find severe tissue damage either histopathologically or by analysis of inflammatory markers. It does however show that MR is very sensitive in detecting potentially damaged regions that may deteriorate over time. The current results also

indicate that future studies on intraluminal pressure should include tissue sample analysis for inflammation in order to examine if higher IRB would elicit a more profound biomarker response or not. Depending on the results of such studies it may also be deducted if IRB in fact is a physiological mechanism that in itself limits the degree of IRP and thus also potential tissue damage.

The most important conclusion of this study is that IRB takes place at even very low pressures, which may alter the assumption that keeping IRP below 30–35 mmHg eliminates the risk of post-operative infection and sepsis. Moreover, the level of IRB was documented to be a function of both IRP and time. The results of our study certainly provide an opportunity to further clarify both short-term and long-term effects on renal function due to the pressure increments in endoscopic procedures The results of this study emphasize the importance of keeping IRP low, although more investigations evaluating the relationship between IRP, OR time and IRB are essential. Hence, further technological improvements and clinical advancements are desirable to secure even better clinical outcomes for patients undergoing ureteroscopic procedures.

## Supporting information

**S1 Table. Histopathology results.** Hematoxylin eosin (H&E) and periodic acid-shiff (PAS) staining of kidney tissue samples. Morphology was evaluated by standard light microscopy. (DOCX)

## Acknowledgments

The authors thank Gitte Kall and Gitte Skou for expert technical laboratory assistance.

## Author Contributions

**Conceptualization:** Søren Kissow Lildal, Esben Søvsø Szocska Hansen, Christoffer Laustsen, Palle Jörn Sloth Osther, Helene Jung.

**Data curation:** Søren Kissow Lildal, Esben Søvsø Szocska Hansen, Christoffer Laustsen, Rikke Nørregaard, Lotte Bonde Bertelsen, Kirsten Madsen, Camilla W. Rasmussen, Helene Jung.

**Formal analysis:** Søren Kissow Lildal, Esben Søvsø Szocska Hansen, Christoffer Laustsen, Rikke Nørregaard, Lotte Bonde Bertelsen, Kirsten Madsen, Helene Jung.

**Investigation:** Søren Kissow Lildal, Esben Søvsø Szocska Hansen, Helene Jung.

**Methodology:** Søren Kissow Lildal, Esben Søvsø Szocska Hansen, Christoffer Laustsen, Rikke Nørregaard, Lotte Bonde Bertelsen, Palle Jörn Sloth Osther, Helene Jung.

**Project administration:** Søren Kissow Lildal, Esben Søvsø Szocska Hansen, Helene Jung.

**Supervision:** Palle Jörn Sloth Osther.

**Visualization:** Esben Søvsø Szocska Hansen.

**Writing – original draft:** Søren Kissow Lildal, Helene Jung.

**Writing – review & editing:** Søren Kissow Lildal, Esben Søvsø Szocska Hansen, Christoffer Laustsen, Rikke Nørregaard, Lotte Bonde Bertelsen, Palle Jörn Sloth Osther, Helene Jung.

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
