## [Decision Letter · Decision Letter 0]

31 Oct 2022

PONE-D-22-27295GADOLINIUM-ENHANCED MRI VISUALIZING BACKFLOW AT INCREASING INTRA-RENAL PRESSURE IN A PORCINE MODELPLOS ONE

Dear Dr. Lildal,

Thank you for submitting your manuscript to PLOS ONE. After careful consideration, we feel that it has merit but does not fully meet PLOS ONE’s publication criteria as it currently stands. Therefore, we invite you to submit a revised version of the manuscript that addresses the points raised during the review process.

We look forward to receiving your revised manuscript.

Kind regards,

Kartikeya Rajdev, MD

Academic Editor

PLOS ONE

Journal Requirements:

"No"

4. Please amend your manuscript to include your abstract after the title page

5. Please ensure that you refer to Figures 6 and 8 in your text as, if accepted, production will need this reference to link the reader to the figure.

6. We note you have included a table to which you do not refer in the text of your manuscript. Please ensure that you refer to Table 1 in your text; if accepted, production will need this reference to link the reader to the Table.

7. Please include a copy of Table 2 which you refer to in your text on page 3. 

Reviewers' comments:

Reviewer's Responses to Questions

**Comments to the Author**

1. Is the manuscript technically sound, and do the data support the conclusions?

Reviewer #1: Yes

Reviewer #2: Yes

2. Has the statistical analysis been performed appropriately and rigorously? 

Reviewer #1: I Don't Know

Reviewer #2: Yes

3. Have the authors made all data underlying the findings in their manuscript fully available?

Reviewer #1: Yes

Reviewer #2: Yes

4. Is the manuscript presented in an intelligible fashion and written in standard English?

Reviewer #1: Yes

Reviewer #2: Yes

5. Review Comments to the Author

Reviewer #1: The study examines the possibility of rare but significant side effect of sepsis with ureteroscopy. The pathophysiology is well explained in the abstract summarized in the following lines " When advancing a ureteroscope to the upper urinary tract and using saline irrigation,

which is necessary for vision, the intrarenal pressure will increase significantly and

when exceeding certain thresholds, intrarenal backflow may occur: backflow of urine and

irrigation fluid to the renal tubules and the venous system."

Reviewer #2: Your study definitely sets stage for further studies to identify safest way to perform urological procedure. Questions like- (1) if short-lasting, extensive pressure increments exert more damage than long-lasting, moderate pressure. (2) MR cab be sensitive in detecting potentially damaged regions that may deteriorate over time but it possibly only imaging changes without any significant clinical outcomes.

I have some questions regarding your study-

You had mentioned no histological signs of inflammation or acute tissue damage could be found and then mentioned it takes some time 2-4weeks to show histopathological signs later in article, Is this time for histopathological changes specific for humans or other animals?

Data showed increased MCP-1 mRNA expression in the treated kidneys compared to contralateral control kidneys. There was no significant change in mRNA expression of the other inflammation markers. It further suggests there may not be significant inflammatory changes in tissue itself and MRI findings are just related to gadolinium contrast entering tissue but may not have actual pathological changes.

IRP during ureteroscopic treatment of kidney stones can reach 3-400 mm Hg[4,5]. Are there any studies measuring IRP with obstructing stones causing hydronephrosis before doing any procedures.

Using this new method, we were able to determine the mean percentage of affected kidney tissue according to IRP and time exposure. The T1 mapping revealed a mean percentage of affected kidney of 66% at a mean IRP of 43 mmHg. Have you monitored reversibility if we stop procedure at certain time and after few minutes/hours MRI shows no residual IRB?

6. PLOS authors have the option to publish the peer review history of their article (what does this mean?). If published, this will include your full peer review and any attached files.

Reviewer #1: **Yes: **Simhachalam Gurugubelli

Reviewer #2: No

---

## [Author Response · Author response to Decision Letter 0]

2 Jan 2023

Response to reviewers

Journal Requirements:

Answer: The manuscript has been changed according to the requirements.

"No"

Answer: The statement: ”The authors have declared that no competing interests exist.” has been added to the cover letter.

Answer: The phrase: ”data not shown” has been removed, as it is not a core part of the study. The histopathology section now merely states that no histological findings were present.

4. Please amend your manuscript to include your abstract after the title page

Answer: The abstract has been added to the manuscript.

5. Please ensure that you refer to Figures 6 and 8 in your text as, if accepted, production will need this reference to link the reader to the figure.

Answer: References have been added to figs 6 and 8.

6. We note you have included a table to which you do not refer in the text of your manuscript. Please ensure that you refer to Table 1 in your text; if accepted, production will need this reference to link the reader to the Table.

Answer: Referral to Figure 2 has been changed to Figure 1. There is no Figure 2. This was a typing error.

7. Please include a copy of Table 2 which you refer to in your text on page 3. 

Answer: Referral to Figure 2 has been changed to Figure 1. There is no Figure 2. This was a typing error.

Reviewers' comments:

Reviewer's Responses to Questions

Comments to the Author

1. Is the manuscript technically sound, and do the data support the conclusions?

Reviewer #1: Yes

Reviewer #2: Yes

2. Has the statistical analysis been performed appropriately and rigorously? 

Reviewer #1: I Don't Know

Reviewer #2: Yes

3. Have the authors made all data underlying the findings in their manuscript fully available?

Reviewer #1: Yes

Reviewer #2: Yes

4. Is the manuscript presented in an intelligible fashion and written in standard English?

Reviewer #1: Yes

Reviewer #2: Yes

5. Review Comments to the Author

Reviewer #1: The study examines the possibility of rare but significant side effect of sepsis with ureteroscopy. The pathophysiology is well explained in the abstract summarized in the following lines " When advancing a ureteroscope to the upper urinary tract and using saline irrigation,

which is necessary for vision, the intrarenal pressure will increase significantly and

when exceeding certain thresholds, intrarenal backflow may occur: backflow of urine and

irrigation fluid to the renal tubules and the venous system."

Answer: Thank you for the comment.

Reviewer #2: Your study definitely sets stage for further studies to identify safest way to perform urological procedure. Questions like- (1) if short-lasting, extensive pressure increments exert more damage than long-lasting, moderate pressure. (2) MR cab be sensitive in detecting potentially damaged regions that may deteriorate over time but it possibly only imaging changes without any significant clinical outcomes.

Answer: Thank you for the interesting comments.

(1) Indeed this is an interesting question. In this study we aimed to show what happens in a setting with relatively low pressure in order to find a threshold for intrarenal backflow. Another study to perform would be a ”worst case scenario” study, similarly performed but with increasing pressures.

(2) We do not know if this MR detected IRB leads to clinically significant damage. This is merely a preliminary study to detect the lower limit of IRB. We do however have plans to perform further studies regarding the potential damages caused by backflow with other MR modalities able to show changes in kidney physiology/metabolism in the future.

I have some questions regarding your study-

You had mentioned no histological signs of inflammation or acute tissue damage could be found and then mentioned it takes some time 2-4weeks to show histopathological signs later in article, Is this time for histopathological changes specific for humans or other animals

Answer: The studies referred to were both done in animal settings with minipigs. Histopathological studies on kidney damages following ureteroscopy are not possible to do in humans as you have to remove the kidney to examine the tissue. We suggested a future study similar to the present study, also in pigs surviving for 2-4 weeks before removing the kidney for examination. Thus we do not know for sure if these earlier findings are specific for pigs but in general pig upper urinary tracts are considered to be very similar to human upper urinary tracts.

Data showed increased MCP-1 mRNA expression in the treated kidneys compared to contralateral control kidneys. There was no significant change in mRNA expression of the other inflammation markers. It further suggests there may not be significant inflammatory changes in tissue itself and MRI findings are just related to gadolinium contrast entering tissue but may not have actual pathological changes.

Answer: This is correct. In this study we can only deduct that the threshold for IRB is very low in vivo as we set out to. We aimed for finding this threshold and thus performed the study in a low pressure setting. The finding of no serious damages is therefore reasonable and we did not expect to find serious damages at least not in short term. The study aimed to simulate what happens generally in human procedures and we do not see very often that kidney function deteriorates significantly shortly after surgery. To find out if it is just imaging changes or if there is a clinically significant damage to kidney physiology we are performing further studies with other MR protocols.

IRP during ureteroscopic treatment of kidney stones can reach 3-400 mm Hg[4,5]. Are there any studies measuring IRP with obstructing stones causing hydronephrosis before doing any procedures.

Answer: Not to our knowledge, and this is also a very difficult study to do. But there are animal studies on partial and complete obstruction showing deterioration of kidney function om MR.

Using this new method, we were able to determine the mean percentage of affected kidney tissue according to IRP and time exposure. The T1 mapping revealed a mean percentage of affected kidney of 66% at a mean IRP of 43 mmHg. Have you monitored reversibility if we stop procedure at certain time and after few minutes/hours MRI shows no residual IRB?

Answer: We have not yet done this. We would expect that the Gadolinium contrast would be washed out and thus the MR changes would be reversible in our opinion. We do not know the time frame for this. We are not planning to examine this using Gadolinium but we are in the proces of examining it using other MR protocols more specific for kidney physiology.

---

## [Decision Letter · Decision Letter 1]

30 Jan 2023

Gadolinium-enhanced MRI visualizing backflow at increasing intra-renal pressure in a porcine model

PONE-D-22-27295R1

Dear Dr. Lildal,

We’re pleased to inform you that your manuscript has been judged scientifically suitable for publication and will be formally accepted for publication once it meets all outstanding technical requirements.

Kind regards,

Kartikeya Rajdev, MD

Academic Editor

PLOS ONE

Reviewers' comments:

Reviewer's Responses to Questions

**Comments to the Author**

1. If the authors have adequately addressed your comments raised in a previous round of review and you feel that this manuscript is now acceptable for publication, you may indicate that here to bypass the “Comments to the Author” section, enter your conflict of interest statement in the “Confidential to Editor” section, and submit your "Accept" recommendation.

Reviewer #2: All comments have been addressed

Reviewer #3: All comments have been addressed

2. Is the manuscript technically sound, and do the data support the conclusions?

Reviewer #2: Yes

Reviewer #3: Yes

3. Has the statistical analysis been performed appropriately and rigorously? 

Reviewer #2: Yes

Reviewer #3: I Don't Know

4. Have the authors made all data underlying the findings in their manuscript fully available?

Reviewer #2: Yes

Reviewer #3: Yes

5. Is the manuscript presented in an intelligible fashion and written in standard English?

Reviewer #2: Yes

Reviewer #3: Yes

6. Review Comments to the Author

Reviewer #2: Overall well written study, it sets a stage for further studies for safe way to perform procedure. Overall data is presented in thorough manner. With limitations of study, you can assume MRI can detect acute damage to kidney at more sensitive way and at earlier time.

Reviewer #3: (No Response)

7. PLOS authors have the option to publish the peer review history of their article (what does this mean?). If published, this will include your full peer review and any attached files.

Reviewer #2: No

Reviewer #3: No

---

## [Editor Report · Acceptance letter]

5 Feb 2023

PONE-D-22-27295R1 

Gadolinium-enhanced MRI visualizing backflow at increasing intra-renal pressure in a porcine model 

Dear Dr. Lildal:

I'm pleased to inform you that your manuscript has been deemed suitable for publication in PLOS ONE. Congratulations! Your manuscript is now with our production department. 

Kind regards, 

on behalf of

Dr. Kartikeya Rajdev 

Academic Editor

PLOS ONE